# Residual Behavior and Dietary Risk Assessment of Chlorfenapyr and Its Metabolites in Radish

**DOI:** 10.3390/molecules28020580

**Published:** 2023-01-06

**Authors:** Mingna Sun, Xiaotong Yi, Zhou Tong, Xu Dong, Yue Chu, Dandan Meng, Jinsheng Duan

**Affiliations:** 1Institute of Plant Protection and Agro-Product Safety, Anhui Academy of Agricultural Sciences, Hefei 230031, China; 2Key Laboratory of Agro-Product Safety Risk Evaluation (Hefei), Ministry of Agriculture, Hefei 230031, China

**Keywords:** chlorfenapyr, UPLC-MS/MS, residual behavior, dietary risk assessment

## Abstract

Chlorfenapyr, as a highly effective and low-toxicity insect growth regulation inhibitor, has been used to control cross-cruciferous vegetable pests. However, the pesticide residue caused by its application threatens human health. In this paper, the residue digestion and final residue of chlorfenapyr in radish were studied in a field experiment. The results of the dynamic digestion test showed that the half-life of chlorfenapyr in radish leaves ranged from 6.0 to 6.4 days, and the digestion rate was fast. The median residual values of chlorfenapyr in radish and radish leaves at 14 days after treatment were 0.12 and 3.92 mg/kg, respectively. The results of the dietary intake risk assessment showed that the national estimated daily intake (NEDI) of chlorfenapyr in various populations in China were 0.373 and 5.66 µg/(kg bw·d), respectively. The risk entropy (RQ) was 0.012 and 0.147, respectively, indicating that the chronic dietary intake risk of chlorfenapyr in radish was low. The results of this study provided data support and a theoretical basis for guiding the scientific use of chlorfenapyr in radish production and evaluating the dietary risk of chlorfenapyr in vegetables.

## 1. Introduction

Chlorfenapyr [1], an aryl-substituted pyrrole compound, interferes with electron transport in the respiratory chain by acting on multifunctional oxidases in insects [2,3] and thus affects energy conversion in insects. Chlorfenapyr mainly kills pests by gastric toxicity, palpation and endosuction, and has a long-lasting effect on lepidoptera pests such as *Diaphorina citri Kuwayama* [4], *Bradysia odoriphaga* [5,6]. In China, chlorfenapyr has been registered and used on crops such as vegetables, fruits, tea trees and ornamental chrysanthemums. The irrational use of pesticides caused by the resistance of insect pests [7,8,9] leads to the increasingly serious residual contamination of chlorfenapyr [10,11,12,13] in vegetables and other agricultural products. This poses a potential threat to non-target organisms [14,15] and human health [16,17]. In addition, chlorfenapyr is also degraded and metabolized in plants to produce a product of toxicological significance, tralopyril. A potency factor of 10 was established for a comparison of the exposure of tralopyril with both the ADI and ARfD of chlorfenapyr at the 2013 Joint FAO/WHO Meeting of Pesticide Residues (JMPR). Although the maximum total radioactive residue of tralopyril is only 3.3%, it may still contribute significantly to the total toxicological burden. Therefore, in the 2018 JMPR, it was recommended that the residues for dietary risk assessment of chlorfenapyr in plant- and animal-derived commodities be defined as the sum of chlorfenapyr and 10-fold tralopyril.

Radish is a popular cruciferous vegetable in the Chinese urban and rural population. Radish is not only nutritious but there also are a variety of ways to eat it. Radish can be cooked or eaten raw, radish leaves can also be stir-fried or used for pickled food. So it is widely grown throughout the country. As a registered pesticide directly acting on the pests of cruciferous vegetables, chlorfenapyr has a good control effect, a long-lasting period and is not easy to degrade. At present, the residue detection of chlorfenapyr in tea [18], chieh-qua [19], pumpkin and okra [20], cabbage [21,22,23], eggplant [24], sweet persimmon [25] and leek [26] has been reported. However, there has been no report on the residual digestion dynamics of chlorfenapyr in radish and its dietary risk assessment for the population. Most of the reported residual detection methods use gas chromatography and liquid chromatography. Moreover, there are few reports on the detection of tralopyril. Liquid chromatography-mass spectrometry/mass spectrometry (LC-MS/MS) [27,28,29,30] and gas chromatography-mass spectrometry/mass spectrometry (GC-MS/MS) [31,32] can not only effectively separate and analyze compounds by chromatography, but also effectively qualitatively analyze substances by mass spectrometry. The complementary advantages of chromatography and mass spectrometry can be used for the qualitative and quantitative analysis of complex organic compounds. They have the characteristics of a wide analysis range, sensitivity and resolution, low dosage and fast analysis speed. The QuEChERS method [33,34,35,36] has been used to monitor many pesticide residues in fruits and vegetables since its introduction, which is fast, simple, cheap and effective.

Therefore, a residue analysis method for chlorfenapyr and its metabolite tralopyril in radish and radish leaves was established. Chlorfenapyr and bromopyrronitrile were detected by GC-MS/MS and ultra-performance liquid chromatography-mass spectrometry/mass spectrometry (UPLC-MS/MS), respectively. The residue behavior of chlorfenapyr on radish and the risk of dietary intake were studied and evaluated by conducting good agricultural practice field residue experiments in six fields. This provided a scientific basis for the rational use of chlorfenapyr and a maximum residue limit (MRL) on radish.

## 2. Results and Discussion

### 2.1. Method Validation

Typical spectra of chlorfenapyr standard in acetonitrile and in radish matrix extract detected by GC-MS/MS are shown in Figure 1 and Figure 2, respectively. The concentration of chlorfenapyr was 0.1 mg/L and the retention time was 8.1 min. The typical spectra of tralopyril in acetonitrile and in the radish leaf matrix extract detected by UPLC-MS/MS are shown in Figure 3 and Figure 4, respectively. The concentration of tralopyril was 0.1 mg/L and the retention time was 0.62 min.

The linearity and fortified recovery of chlorfenapyr and tralopyril in radish and radish leaf matrix are shown in Table 1. There was a good linear relationship between the peak area of chlorfenapyr and the injection concentration in the range of 0.01–1.0 mg/L and 0.01–12 mg/L on the radish and radish leaf matrix, respectively. R^2^ was 0.9974 and 0.9948, respectively. The LOQ of chlorfenapyr in radish and radish leaf matrix was 0.01 mg/kg. The average recoveries of chlorfenapyr in radish were 88.7% to 95.0%, and the RSD was 2.5% to 3.7%. The average recoveries of chlorfenapyr in radish leaf were 74.0% to 88.3%, and the RSD was 1.7% to 4.4%. The peak area of tralopyril on the radish and radish leaf matrix both showed a good linear relationship with the injection concentration in the range of 0.01–1.0 mg/L. R^2^ were 0.9930 and 0.9936, respectively. The LOQ of chlorfenapyr in radish and radish leaf matrix was 0.01 mg/kg. The average recoveries of tralopyril in radish were 96.9% to 106.3%, and the RSD was 2.4% to 2.7%. The average recoveries of chlorfenapyr in radish leaf were 91.7% to 106.8%, and the RSD was 2.0% to 6.9%. The accuracy and precision of the method meet the requirement of pesticide residue analysis.

### 2.2. Degradation of Chlorfenapyr in Radish Leaf

The residual amount of chlorfenapyr on the radish leaf gradually decreased with time, and its degradation process was in line with the first-order kinetic equation. The specific results were shown in Table 2. The digestion rules of different fields were basically the same: the initial deposition amount of chlorfenapyr in radish leaves was 16.89–17.42 mg/kg. After 21 days of application, the residual digestion rate was more than 90%, and the half-life was 6.0–6.4 days. The degradation was fast. This is similar to the degradation half-life of chlorfenapyr in leafy vegetables reported by others [22,23].

### 2.3. Final Residue Test

The final residual results of chlorfenapyr and tralopyril on radish and radish leaves after treatment with different doses of chlorfenapyr suspension were shown in Table 3 and Table 4. The metabolite tralopyril was detected in radish. However, for radish leaves, no tralopyril was detected in the Chongqing samples; it was detected in the other five samples, and the highest amount was detected in the Guangdong samples.

The final residue test results on radish are as follows: at the last application and harvest interval of 7 days, the residual amounts of chlorfenapyr ranged from 0.067 to 0.48 mg/kg, and the median residues were 0.17 mg/kg; at the last application and harvest interval of 14 days, the residual amounts of chlorfenapyr ranged from 0.025 to 0.42 mg/kg, and the median residues were 0.12 mg/kg; at the last application and harvest interval of 21 days, the residual amounts of chlorfenapyr ranged from 0.022 to 0.12 mg/kg, and the median residues were 0.12 mg/kg. For radish leaves: at the last application and harvest interval of 7 days, the residual amounts of chlorfenapyr ranged from 1.35 to 12.25 mg/kg, and the median residues were 5.99 mg/kg; at the last application and harvest interval of 14 days, the residual amounts of chlorfenapyr ranged from 1.09 to 7.24 mg/kg, and the median residues were 3.92 mg/kg; at the last application and harvest interval of 21 days, the residual amounts of chlorfenapyr ranged from 0.36 to 5.08 mg/kg, respectively, and the median residues were 2.41 mg/kg.

### 2.4. Chronic Dietary Intake Risk Assessment

Considering the long growth cycle of radish, the median residual value of 0.12 mg/kg during the 14-day harvest interval was used to evaluate the long-term dietary risk of chlorfenapyr in radish. The ADI value of chlorfenapyr was 0.03 mg/kg bw. According to the China Health and Nutrition Survey summary report, the daily intake of all light color vegetables (radish belongs to light color vegetables) in Chinese adults is 0.202 kg. The national estimated daily intake (NEDI) of chlorfenapyr in the general population was calculated as 0.373 µg/(kg bw·d) according to Equation (2), and the risk quotient (RQ) was 0.012, less than 1. The median residue of radish leaves during the 14-day harvest interval was 3.92 mg/kg, and the daily intake of all dark vegetables (radish leaves are dark vegetables) in Chinese adults was 0.0937 kg. Similarly, the national estimated daily intake (NEDI) value of chlorfenapyr in the general population was 5.66 µg/(kg bw·d), and the risk quotient (RQ) value was 0.147, which was also less than 1. These results indicated that the dietary exposure risk of chlorfenapyr in radish and radish leaves was low and acceptable.

## 3. Experimental Materials and Methods

### 3.1. Experimental Materials

Chlorfenapyr standards (98.6%) were purchased from Tan-Mo Technology Co., Ltd., Changzhou, China. Tralopyril standards (98.6%) were purchased from Dr. Ehrenstorfer GmbH, Germany. Sodium chloride (AR) was purchased from Sinopharm Chemical Reagent Co., Ltd., Shanghai, Chian. PSA and GCB were purchased from Tianjin Bonaijer Technology Co., Ltd., Tianjin, China. Anhydrous sodium sulfate (AR) was purchased from Shanghai Runjie Chemical Reagent Co., Ltd., Shanghai, China. Chromatographically pure n-hexane and acetonitrile were purchased from Tedia Co., Ltd., Fairfield, USA. The Chlorfenapyr standard and Tralopyril standard were dissolved in n-hexane and acetonitrile to prepare a stock solution with a concentration of 1000 mg/L, stored at 4 °C in the dark. Diluted to different concentrations when used.

### 3.2. Instrument Condition

Chlorfenapyr in radish was tested using a Shimadzu GC-MS/MS-TQ8040 (Shimadzu Corporation, Tokyo, Japan) equipped with an electrospray ion source (ESI). The Rtx-5MS (30 m × 0.25 mm × 0.25 μm) column was used to separate the target compound. The injection volume was 3 μL. The injection method was no shunt. The temperature of the injection port was 250 °C. Gradient temperature: The initial temperature was 60 °C, then increased to 150 °C at 50 °C/min, then increased to 250 °C at 20 °C/min, held for 5 min, then increased to 280 °C at 30 °C/min, held for 1 min. The carrier gas was helium with purity > 99.99% and the column flow rate was 1.5 mL/min. Mass spectrometry conditions were as follows. Multi reaction monitoring (MRM) mode was adopted. The ion source temperature was 230 °C. The interface temperature was 280 °C. The colliding gas was argon with a purity of 99.99%. The capillary voltage was 2 kV. The parent ion was *m*/*z* 247.1. The daughter ions were *m*/*z* 227 (quantitative) and *m*/*z* 200 and the collision energy was 16 eV and 24 eV.

Tralopyril in radish was tested using a Shimadzu UPLC-MS/MS-8030 (Shimadzu Corporation, Tokyo, Japan) equipped with an electrospray ion source (ESI). Liquid chromatography conditions: the Shim-pack XR-ODSIII column (1.6 μm, 50 mm × 2.0 mm (inside diameter)) was used; mobile phase A was water containing 0.1% formic acid and B was acetonitrile; the flow rate was 0.3 mL/min; the column temperature was set to 40 °C; the injection volume was 10 μL. the gradient elution procedure was as follows: 0–0.01 min, 90% B phase; 0.1–2 min, 90% B–50% B phase; 2–4 min, 50% B phase–90% B phase; 4–5 min, 90% B phase. Mass spectrometry conditions were as follows: ESI mass spectra were recorded in negative ion mode. The ion source temperature was 200 °C. The interface temperature was 350 °C. The DL temperature was 200 °C. The collision gas (CID) was 230 KPa. The parent ion was *m*/*z* 349.0 and the daughter ions were *m*/*z* 131.1 (quantitative) and *m*/*z* 81.1 and the collision energy was 16 eV and 24 eV. 

### 3.3. Design of Field Trial Tests

Field trial tests were carried out in Wuhu of Anhui Province, Shuangliao of Jilin Province, Weifang of Shandong Province, Wuhan of Hubei Province, Jiulongpo of Chongqing Municipality and Zhaoqing of Guangdong Province in 2019. The experiment was carried out in accordance with the requirements of “Guidelines for Pesticide Residue Testing” (NY/T788-2004) and “Standard Operating Procedures for Pesticide Registration Residue Field Experiments”. Plots were 30 m^2^ in size in triplicate and randomly arranged. Protection belts were set between the plots. Another control plot was set up. The recommended dosage of 240 g/L chlorfenapyr suspension for the control of *Plutella xylostella* and *Beet armyworm* on radish was 90–108 g a.i./ha (formulation volume was 25–30 mL/667 m^2^). The insecticide was applied twice at the initial stage of insect infestation with an interval of 7 days.

#### 3.3.1. Dynamic Digestion Test

In Anhui and Jilin, the dose of 162 g a.i./ha (45 mL/667 m^2^, 1.5 times the recommended high dose) was applied once at the mid-growth stage of radish, and the radish leaves were evenly sprayed. The radish leaves were collected at 2 h, 1 d, 3 d, 5 d, 7 d, 14 d, 21 d, and 30 d after application. In the experimental plot, 12 or more aboveground leaves were randomly collected, and three to six representative leaves were collected from each plant according to the inner, middle and outer parts. At least 1000 g samples were taken from each plot, finely chopped and thoroughly mixed, divided into four fractions, and 200 g samples were put into the sample sack with three replicates. They were labeled inside and outside and stored in the refrigerator at −20 °C until the test. At the same time, blank control radish leaf samples were collected.

#### 3.3.2. Final Residue Test

The final residue test was carried out in the six fields mentioned above. At the early stage of radish growth, two or three doses of 108 g a.i./ha (dosage of 30 mL/667 m^2^, the recommended high dose) and 162 g a.i./ha (dosage of 45 mL/667 m^2^, 1.5 times of the recommended high dose) were applied, and the application interval was 7 days. Samples of radish and radish leaves were collected at intervals of 7 d, 14 d, and 21 d after the last application. The sampling method of radish leaves was the same as the dynamic digestion test. At the maturity stage of radish growth, samples were collected randomly in the experimental plot; 6–12 radish samples (depending on the size of the radish) were collected from each plot, with at least 2000 g samples each time. We removed the surface soil, wiped them dry and minced and mixed well. We reduced fractions by the rule of four; 200 g samples were put into the sample bag, and the samples were repeated in triplicate. We labeled the container inside and outside and stored it in the refrigerator at −20 °C until the test. A blank control radish and radish leaf samples were collected.

### 3.4. Sample Pretreatment

More than 20 g of samples were broken with a homogenizer, and 5 g samples were accurately weighed and put into a 50 mL centrifuge tube; 10 mL of acetonitrile was added. The samples were vortexed at 1800 rpm for 2 min, then we added 2 g of NaCl, and centrifuged at 5000 rpm for 3 min. The samples were left to stand until the acetonitrile extract and NaCl solution was split into different layers; 5 mL of supernatant acetonitrile extract was removed into a 10 mL centrifuge tube (equipped with 200 mg PSA and 30 mg GCB), vortexed at 1200 rpm for 1 min; 2 mL of the purified acetonitrile extract was transferred through a 0.22 μm organic filter membrane for the determination of tralopyril by UPLC-MS/MS. Then 2 mL of the purified acetonitrile extract was transferred to a 150 mL concentration bottle, concentrated to near dry in a 40 °C water bath, and fixed with 2 mL n-hexane (chromatographic grade). We used GC-MS/MS to determine the chlorfenapyr after passing it through a 0.22 μm organic filter membrane.

### 3.5. Method Validation

The blank radish and blank radish leaves were supplemented with chlorfenapyr and tralopyril standards; five parallel tests were performed for each concentration. Extraction and purification were carried out according to method 3.4. Detection was carried out according to detection condition 3.2. The recovery rate and relative standard deviation were measured. An appropriate amount of standard stock solution was accurately removed and diluted with blank radish and blank radish leaf matrix extract, respectively. We prepared 0.005, 0.01, 0.05, 0.1, 0.5, and 1.0 mg/L chlorfenapyr and tralopyril matrix standard solutions for sample quantification.

### 3.6. Data Statistics

The assessed and defined residues of chlorfenapyr were calculated according to Equation (1):(1)$$X=C_{1}+10\times C_{2}$$

According to the assessment definition, the residue of the dietary assessment of chlorfenapyr was defined as the sum of chlorfenapyr and the 10 times metabolite tralopyril. Therefore, *X* was the residual amount of chlorfenapyr (mg/kg) in the sample.

The risk of chronic dietary exposure was calculated according to Equations (2) and (3):(2)NEDI=(∑STMRi×Fi)/bw
(3)RQ=NEDI/ADI

In Equation (2), *NEDI* is the estimated daily intake per capita in China, μg/(kg bw·d). *STMRi* refers to the median residual value of grade *i* agricultural products in a standardized test, mg/kg. *Fi* is the dietary consumption of grade *i* agricultural products in different populations, and the unit is g.

In Equation (3), *RQ* is the risk entropy, and *ADI* is the daily allowable intake of pesticides per kilogram of body weight, mg/kg bw. When *RQ* ≤ 1, the risk is acceptable. The smaller *RQ* is, the lower the risk is. When *RQ* > 1, it indicates an unacceptable chronic risk. A larger *RQ* indicates a greater risk.

## 4. Conclusions

In this study, a method for the residue analysis of chlorfenapyr and its metabolite tralopyril, on radish and radish leaves was established, and the residue samples of chlorfenapyr in six fields were detected by the method. The results showed that the degradation rate of chlorfenapyr in radish leaves was fast, and the degradation half-life was from 6.0 to 6.4 days, indicating that chlorfenapyr was an easily degradable pesticide. Only a small amount of tralopyril was detected in radish leaves. When the application dose of 240 g/L chlorfenapyr suspension was 108 and 162 g a.i./ha and the spray was applied two to three times with an interval of 7 days, the residual chlorfenapyr in radish and radish leaf samples collected 14 days after the last application had little risk for the health of the general population in China. At present, China’s national food safety standard GB2763-2021 has not set the maximum residual limit of chlorfenapyr on radish, which is 2 mg/kg in the United States, 0.1 mg/kg in radish and 3 mg/kg in radish leaves in Japan. The results of this study provide a data basis for establishing the maximum residual limit of chlorfenapyr in radishes in China.

## Figures and Tables

**Figure 1 molecules-28-00580-f001:**
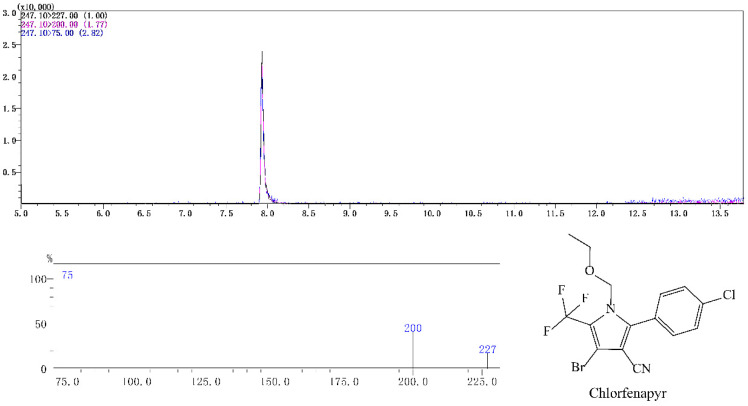
Typical spectra of chlorfenapyr standard in acetonitrile detected by GC-MS/MS. The concentration of chlorfenapyr was 0.1 mg/L.

**Figure 2 molecules-28-00580-f002:**
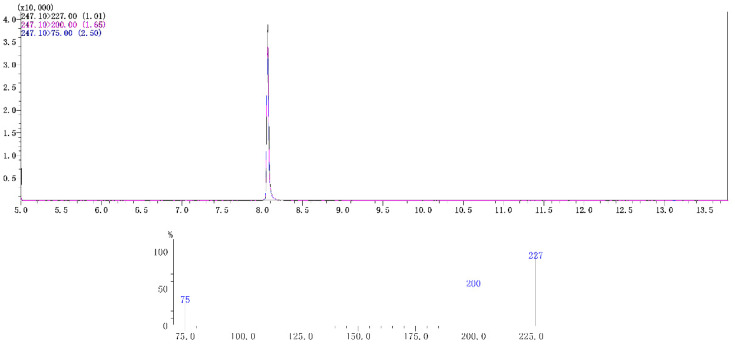
Typical spectra of chlorfenapyr in radish matrix extract detected by GC-MS/MS. The concentration of chlorfenapyr was 0.1 mg/L.

**Figure 3 molecules-28-00580-f003:**
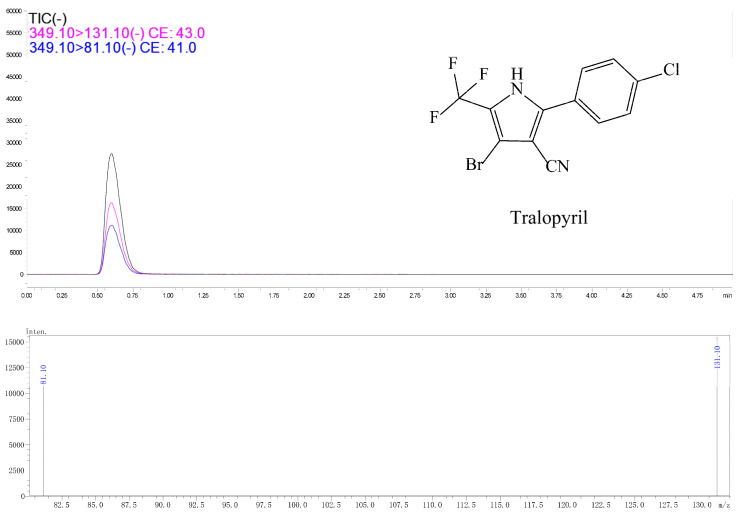
Typical spectra of tralopyril standard in acetonitrile detected by UPLC-MS/MS. The concentration of tralopyril was 0.1 mg/L.

**Figure 4 molecules-28-00580-f004:**
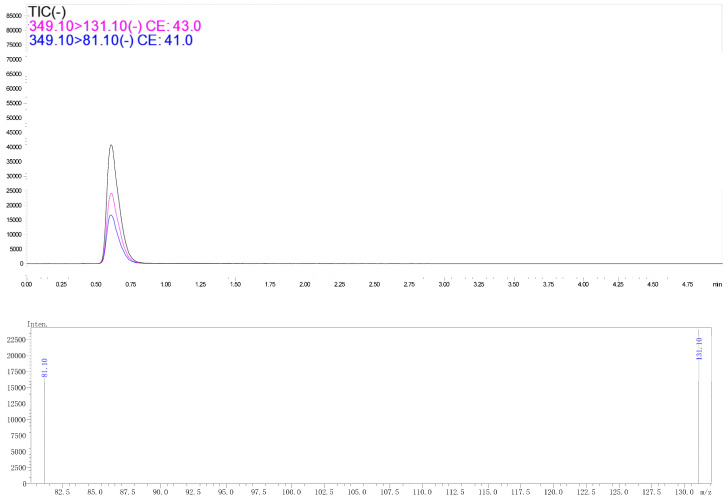
Typical spectra of tralopyril in radish leaf matrix extract detected by UPLC-MS/MS. The concentration of tralopyril was 0.1 mg/L.

**Table 1 molecules-28-00580-t001:** Linear range, regression equation, correlation coefficient (R^2^), fortified recovery, relative standard deviation (RSD) and limit of quantitation (LOD) of chlorfenapyr and tralopyril in radish and radish leaf.

Analyte	Matrix	Linearity Range (mg/L)	Regression Equation	R^2^	Fortified Concentration (mg/kg)	Mean Recovery (%)	RSD (%)	LOQ (mg/kg)
Chlorfenapyr	Radish	0.01–1	Y = 740431.3X − 6894.6	0.9974	0.01	95.0	3.0	0.01
					0.1	88.9	3.7	
					1	88.7	2.5	
	Radish leaf	0.01–12	Y = 1079662.2X − 20821.1	0.9948	0.1	88.3	1.7	0.01
					1	85.3	4.4	
					10	74.0	2.1	
					12	75.8	3.2	
Tralopyril	Radish	0.01–1	Y = 883612.9X + 44608.7	0.9930	0.01	106.3	2.4	0.01
					0.1	101.3	2.8	
					1	96.9	2.7	
	Radish leaf	0.01–1	Y = 846318.0X + 42024.3	0.9936	0.01	91.7	2.0	0.01
					0.1	106.8	4.4	
					1	103.0	6.9	

**Table 2 molecules-28-00580-t002:** Degradation of chlorfenapyr in radish leaf.

Time	Anhui				Huibei			
	Residual Amount (mg/kg)			Digestion Rate (%)	Residual Amount (mg/kg)			Digestion Rate (%)
	Chlorfenapyr	Tralopyril	Total Quantity		Chlorfenapyr	Tralopyril	Total Quantity	
2 h	16.88	<0.01	16.880	—	17.41	<0.01	17.42	—
1 d	14.99	<0.01	15.00	11.14	16.07	<0.01	16.08	2.47
3 d	14.01	0.013	14.14	16.23	16.96	<0.01	16.99	7.70
5 d	7.99	0.019	8.18	51.53	8.91	<0.01	8.93	48.77
7 d	7.45	0.022	7.67	54.59	8.32	0.019	8.52	51.15
14 d	5.96	0.021	6.17	63.44	5.57	<0.01	5.58	67.96
21 d	1.07	<0.01	1.08	93.62	1.33	<0.01	1.34	92.31
30 d	0.76	<0.01	0.77	95.43	0.59	<0.01	0.60	96.58
Equation	y = 17.1518e^−0.1083x^				y = 19.3711e^−0.1162x^			
R^2^	0.9710				0.9876			
T_1/2_	6.4 d				6.0 d			

**Table 3 molecules-28-00580-t003:** Final residues of chlorfenapyr and tralopyril in radish.

Field	Dose/(g a.i./hm^2^)	Application Number	Chlorfenapyr			Tralopyril			Chlorfenapyr (Evaluate Definition)		
			7 d	14 d	21 d	7 d	14 d	21 d	7 d	14 d	21 d
Anhui	108	2	0.057	0.045	0.042	<0.01	<0.01	<0.01	0.067	0.055	0.052
		3	0.113	0.081	0.060	<0.01	<0.01	<0.01	0.123	0.091	0.070
	162	2	0.071	0.060	0.046	<0.01	<0.01	<0.01	0.081	0.070	0.056
		3	0.113	0.106	0.077	<0.01	<0.01	<0.01	0.123	0.116	0.087
Jilin	108	2	0.094	0.084	0.078	<0.01	<0.01	<0.01	0.104	0.094	0.088
		3	0.104	0.083	0.070	<0.01	<0.01	<0.01	0.114	0.093	0.080
	162	2	0.211	0.133	0.097	<0.01	<0.01	<0.01	0.221	0.143	0.107
		3	0.217	0.143	0.111	<0.01	<0.01	<0.01	0.227	0.153	0.121
Hubei	108	2	0.076	0.015	0.012	<0.01	<0.01	<0.01	0.086	0.025	0.022
		3	0.099	0.078	0.012	<0.01	<0.01	<0.01	0.109	0.088	0.022
	162	2	0.084	0.026	0.024	<0.01	<0.01	<0.01	0.094	0.036	0.034
		3	0.159	0.050	0.020	<0.01	<0.01	<0.01	0.169	0.060	0.030
Shandong	108	2	0.104	0.080	0.056	<0.01	<0.01	<0.01	0.114	0.090	0.066
		3	0.299	0.211	0.101	<0.01	<0.01	<0.01	0.309	0.221	0.111
	162	2	0.242	0.090	0.090	<0.01	<0.01	<0.01	0.252	0.100	0.100
		3	0.250	0.159	0.156	<0.01	<0.01	<0.01	0.260	0.169	0.166
Chongqing	108	2	0.130	0.132	0.058	<0.01	<0.01	<0.01	0.140	0.142	0.068
		3	0.254	0.193	0.202	<0.01	<0.01	<0.01	0.264	0.203	0.212
	162	2	0.313	0.281	0.177	<0.01	<0.01	<0.01	0.323	0.291	0.187
		3	0.438	0.400	0.329	<0.01	<0.01	<0.01	0.448	0.410	0.339
Guangdong	108	2	0.125	0.108	0.107	<0.01	<0.01	<0.01	0.135	0.118	0.117
		3	0.199	0.151	0.115	<0.01	<0.01	<0.01	0.209	0.161	0.125
	162	2	0.438	0.119	0.116	<0.01	<0.01	<0.01	0.448	0.129	0.126
		3	0.236	0.104	0.046	<0.01	<0.01	<0.01	0.246	0.114	0.056

**Table 4 molecules-28-00580-t004:** Final residues of chlorfenapyr and tralopyril in radish leaves.

Field	Dose/(g a.i./hm^2^)	Application Number	Chlorfenapyr			Tralopyril			Chlorfenapyr (Evaluate Definition)		
			7 d	14 d	21 d	7 d	14 d	21 d	7 d	14 d	21 d
Anhui	108	2	2.674	2.110	1.684	0.027	0.020	<0.01	2.946	2.311	1.694
		3	3.528	2.545	1.889	<0.01	0.012	<0.01	3.562	2.666	1.899
	162	2	10.60	3.666	2.674	0.026	<0.01	0.013	10.86	3.766	2.800
		3	11.88	5.858	4.984	0.027	<0.01	<0.01	12.14	5.868	4.994
Jilin	108	2	7.438	3.523	2.030	0.013	0.023	0.016	7.571	3.755	2.190
		3	9.640	5.088	2.937	0.035	0.056	0.045	9.994	5.646	3.385
	162	2	10.41	4.627	5.138	0.034	0.053	0.064	10.76	5.156	5.773
		3	10.91	6.690	2.057	0.025	0.027	<0.01	11.23	6.939	2.328
Hubei	108	2	7.501	3.288	1.640	<0.01	<0.01	<0.01	7.511	3.298	1.65
		3	8.012	4.607	2.378	0.012	0.0.15	0.018	8.131	4.756	2.558
	162	2	7.271	5.694	3.369	0.011	0.011	0.014	7.385	5.803	3.509
		3	10.98	4.168	3.148	0.021	0.015	0.016	11.19	4.321	3.421
Shandong	108	2	4.361	3.148	2.063	0.042	0.040	0.028	4.779	3.55	2.343
		3	3.528	3.365	0.925	0.033	0.039	0.027	3.963	3.759	1.195
	162	2	5.275	4.234	2.128	0.023	0.035	0.030	5.505	4.58	2.427
		3	3.846	3.035	1.355	0.026	0.048	<0.01	5.125	4.106	3.51
Chongqing	108	2	1.355	1.092	0.490	<0.01	<0.01	<0.01	1.365	1.102	0.500
		3	4.724	1.606	0.356	<0.01	<0.01	<0.01	4.734	1.616	0.366
	162	2	4.017	2.426	1.490	<0.01	<0.01	<0.01	4.027	2.436	1.500
		3	7.601	4.246	2.239	<0.01	<0.01	<0.01	7.611	4.256	2.249
Guangdong	108	2	4.245	2.225	1.448	0.175	0.110	0.069	5.992	3.328	2.135
		3	3.176	2.413	1.066	0.108	0.086	0.067	4.259	3.270	1.736
	162	2	4.103	3.070	2.791	0.149	0.128	0.097	5.593	4.35	3.758
		3	5.601	3.630	1.690	0.118	0.1240	0.087	6.781	4.873	2.563

## Data Availability

No new data were created or analyzed in this study. Data sharing is not applicable to this article.

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
