# Peer review of "Residual Behavior and Dietary Risk Assessment of Chlorfenapyr and Its Metabolites in Radish"

_molecules, 2023, doi:10.3390/molecules28020580_

Round 1
Reviewer 1 Report
line 99: Add mass spectrometry conditions: the flow rate and pressure of nitrogen, the capillary voltage, collision energy, isolation width. ESI mass spectra were recorded in positive or negative ion mode. What internal standard were used?
line 101 and 102: Designation is missing in parent ions and daughter ions, you mean [M+H]+ or m/z or.....
line 178: The chapter title is wrong.
line 206: You mean....The metobolite of chlorfenapyr (tralopyril) were detected in radish. Or exists other metabolites of tralopyril or chlorfenapyr? I don´t get the sentence.
References 18 and 19: References must be changed, cannot be searched.
Author Response
line 99: Add mass spectrometry conditions: the flow rate and pressure of nitrogen, the capillary voltage, collision energy, isolation width. ESI mass spectra were recorded in positive or negative ion mode. What internal standard were used?
Answer: Thank you for your comments. The mass spectrometry conditions have been supplemented, please see line 102-106 for details. This method don’ t use internal standard for quantification, so there is no relevant description.
line 101 and 102: Designation is missing in parent ions and daughter ions, you mean [M+H]+ or m/z or.....
Answer: Sorry for the mistake. We have added to it in the manuscript. Please see line 105-106 and line 116-117.
line 168: The chapter title is wrong.
Answer: The chapter title has been modified, please see line 172.
line 216: You mean....The metobolite of chlorfenapyr (tralopyril) were detected in radish. Or exists other metabolites of tralopyril or chlorfenapyr? I don´t get the sentence.
Answer: The metabolite is tralopyril. No other metabolites were detected. We have corrected the wrong statement, please see line 222.
References 18 and 19: References must be changed, cannot be searched.
Answer: References 18 and 19 are JMPR reports and have been removed.
Reviewer 2 Report
Analytical work is done correctly. Chromatographic analyzes are well designed. But there is no information about the substances analysed.Authors must include chromatograms and mass spectra: 1. standards for GC/MSMS and HPLC/MSMS 2. include exemplary chromatograms from the tested samples with a description. Then the work will be complete and will be helpful for other researchers in terms of methodology.
Author Response
Analytical work is done correctly. Chromatographic analyzes are well designed. But there is no information about the substances analysed.
Authors must include chromatograms and mass spectra: 1. standards for GC/MSMS and HPLC/MSMS 2. include exemplary chromatograms from the tested samples with a description. Then the work will be complete and will be helpful for other researchers in terms of methodology.
Answer: Thank you for your comments. We have added the typical spectra of GC-MS/MS detection of chlorfenapyr in radish matrix standard solution and UPLC-MS/MS detection of tralopyril in radish leaf matrix standard solution, as shown in Figure 2 and lines 192-196.
Round 2
Reviewer 2 Report
In my first opinon i ask for:
"Authors must include chromatograms and mass spectra:
1. standards for GC/MSMS and HPLC/MSMS
2. include exemplary chromatograms from the tested samples with a description.
Then the work will be complete and will be helpful for other researchers in terms of methodology."
The authors did not follow the comments from the previous version.
Author Response
Thank you for your comments. Please see the attachment for corresponding reply.
